# Signal and noise in metabarcoding data

Zachary Gold[1,2]*, Andrew Olaf Shelton[2], Helen R. Casendino[3], Joe Duprey[3], Ramón Gallego[2], Amy Van Cise[2], Mary Fisher[4], Alexander J. Jensen[2], Erin D'Agnese[3], Elizabeth Andruszkiewicz Allan[3], Ana Ramón-Laca[2], Maya Garber-Yonts[3], Michaela Labare[5], Kim M. Parsons[2], Ryan P. Kelly[3]

**1** Cooperative Institute for Climate, Ocean, & Ecosystem Studies, UW, Seattle, Washington, United States of America, **2** Northwest Fisheries Science Center, NMFS/NOAA, Seattle, Washington, United States of America, **3** School of Marine and Environmental Affairs, UW, Seattle, Washington, United States of America, **4** School of Aquatic Fisheries Science, UW, Seattle, Washington, United States of America, **5** Scripps Institution of Oceanography, UCSD, La Jolla, California, United States of America

* Zachary.gold@noaa.gov

**Data Availability Statement:** All data and code for conducting analyses are publicly available via NCBI SRA (BioProject: PRJNA966238), Dryad (https://doi.org/10.5068/D1XH5H), and GitHub (https://doi.org/10.5281/zenodo.7887383).

## Abstract

Metabarcoding is a powerful molecular tool for simultaneously surveying hundreds to thousands of species from a single sample, underpinning microbiome and environmental DNA (eDNA) methods. Deriving quantitative estimates of underlying biological communities from metabarcoding is critical for enhancing the utility of such approaches for health and conservation. Recent work has demonstrated that correcting for amplification biases in genetic metabarcoding data can yield quantitative estimates of template DNA concentrations. However, a major source of uncertainty in metabarcoding data stems from non-detections across technical PCR replicates where one replicate fails to detect a species observed in other replicates. Such non-detections are a special case of variability among technical replicates in metabarcoding data. While many sampling and amplification processes underlie observed variation in metabarcoding data, understanding the causes of non-detections is an important step in distinguishing signal from noise in metabarcoding studies. Here, we use both simulated and empirical data to 1) suggest how non-detections may arise in metabarcoding data, 2) outline steps to recognize uninformative data in practice, and 3) identify the conditions under which amplicon sequence data can reliably detect underlying biological signals. We show with both simulations and empirical data that, for a given species, the rate of non-detections among technical replicates is a function of both the template DNA concentration and species-specific amplification efficiency. Consequently, we conclude metabarcoding datasets are strongly affected by (1) deterministic amplification biases during PCR and (2) stochastic sampling of amplicons during sequencing—both of which we can model—but also by (3) stochastic sampling of rare molecules prior to PCR, which remains a frontier for quantitative metabarcoding. Our results highlight the importance of estimating species-specific amplification efficiencies and critically evaluating patterns of non-detection in metabarcoding datasets to better distinguish environmental signal from the noise inherent in molecular detections of rare targets.

**Funding:** ZG was supported by the Joint Institute for the Study of the Atmosphere and Ocean (JISAO) under NOAA Cooperative Agreement NA15OAR4320063. AJJ was supported by NOAA and University of Washington. RPK was supported by the David and Lucile Packard Foundation. EAA and ED were supported by OceanKind. The funders had no role in study design, data collection and analysis, decision to publish, or preparation of the manuscript.

**Competing interests:** Authors have no conflicts of interest to report.

## Introduction

Metabarcoding, or DNA amplicon sequencing, is a powerful tool that can characterize biological communities without the need to physically observe individual organisms. This biological monitoring tool utilizes polymerase chain reaction (PCR) to target conserved DNA gene regions and subsequently characterize hundreds or thousands of species from a given sample [1–4]. Over the past twenty years, the rise of metabarcoding via high-throughput sequencing has rapidly advanced human and wildlife health, ecology, and conservation science allowing for the characterization of microbiomes, environmental DNA, and gut content analyses among many other applications [2–14]. Increasingly, the results of metabarcoding data influence healthcare and conservation management and policy decisions [15, 16], and therefore it is increasingly important to improve our interpretation of metabarcoding data.

In particular, the (largely unrealized) power of metabarcoding applications lies in the ability to obtain reliable quantitative estimates of underlying communities [17–19]. In the case of metabarcoding and similar amplicon-based studies [20], it has become clear that 1) observations are non-linearly related to the underlying biology of interest [21, 22], and 2) those observations are noisy, with many having relatively high variances as a function of expected values [19, 23–25]. To obtain reliable quantitative estimates for any set of observations, we must be able to distinguish random variation from real signal. Thus, understanding the underlying signal-to-noise ratio is key to quantifying the power of species-level detection–and accuracy of quantification–in a given dataset [26].

Substantial efforts to correlate sequence reads and underlying community abundance have reported promising but largely equivocal results [18, 23, 27–31]. However, it is unsurprising that the application of simple linear correlations to non-linear and compositional datasets produce ambiguous results given the failure to model the underlying drivers of observed DNA sequence patterns and distributions. In response, recent mechanistic frameworks have begun to address the discrepancies between observed metabarcoding sequence counts and true underlying biological patterns by modeling the compounding processes that occur between DNA extraction and sequence observation [32–36]. These processes include DNA extraction, PCR, and multiple subsampling steps prior to sequencing [24, 25, 35, 37, 38]. We model the collection process after Shelton et al. [37] (Fig 1).

Importantly, a suite of mechanistic frameworks explicitly model the amplicon sequence-generating process by stating that observed sequence reads are a function of both the species-specific amplification efficiency and the underlying abundance of each species' DNA within a sample [34]. Such models also reflect the inherent compositional nature of metabarcoding, acknowledging that metabarcoding data can only provide proportional (not absolute) abundances of a given species' DNA in each sample [19]. This approach can reconstruct starting DNA proportions, prior to PCR (e.g. [24, 33–35]) and, if metabarcoding data are combined with additional information on underlying DNA concentrations (e.g., via qPCR), can yield absolute abundance estimates (e.g. [39]).

Despite these advances in modeling the amplicon sequence-generating process, it is clear that the sequential molecular steps required to generate metabarcoding data will result in variable sequence-read counts among technical replicates derived from the same DNA extract [21, 39–43]. Thus, in practice, it can be difficult to distinguish signal from noise in metabarcoding datasets. In particular, zeros or non-detections (in which a species is unobserved in one technical replicate despite being observed in other replicates) are frequently over-represented in metabarcoding data, contributing substantially to among-replicate variability [24, 32, 40]. For example, in three technical replicates, a unique amplicon sequence variant (ASV) may be represented by 3,897; 165; and 0 reads across replicates (132,731, 196,260, 55,400 read depth for

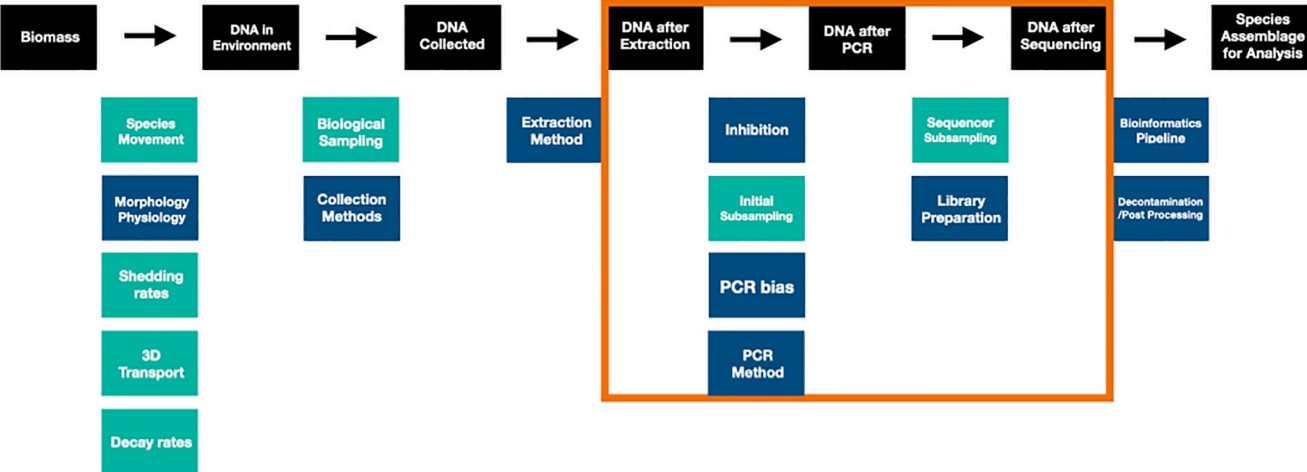

**Fig 1. Compounding processes of metabarcoding that affect observed sequence patterns.** Observed sequences from metabarcoding are impacted by a suite of deterministic (navy) and stochastic (teal) processes. Here we focus on modeling the processes between extracted DNA and observed DNA sequences highlighted within the orange box.

each replicate respectively; [39]). This observed variability among technical replicates far exceeds the expected variability arising from binomial- or multinomial sampling, and so demands a different explanation [24].

Here we focus on the patterns and causes of non-detections in metabarcoding datasets. We specifically build on the metabarcoding framework proposed in Shelton et al. [35], in which species-specific amplification efficiencies strongly influence observed sequence proportions, and additionally explore the effects of subsampling rare molecules prior to PCR amplification on patterns of sequence counts and non-detections. Previous work has explored the effects of amplification efficiencies and subsampling on observed metabarcoding results individually [24, 25, 33, 34], but no study to date has modeled both processes simultaneously to explore their interactive and relative effects on observed sequence data.

To address this, we simulate a stochastic subsampling process from the DNA extraction prior to the PCR reaction, a deterministic PCR process, and a stochastic subsampling process during DNA sequencing to develop a qualitative understanding of the scenarios under which non-detections arise. We then use these results to generate predictions for the frequency of non-detections. Finally, we use empirical observations to test these predictions using metabarcoding data derived from a set of ethanol-preserved fish larvae [39], in which both the underlying organismal abundances and the resulting metabarcoding dataset are well-characterized. Our empirical findings closely match the predictions from simulations and suggest that both mechanisms (subsampling and amplification bias) contribute to non-detections and stochastic variability in metabarcoding data. Given this understanding of the sources of variability, we can more confidently distinguish signal from noise in metabarcoding datasets.

## Methods

### Conceptual model and simulating metabarcoding data

Our generating model for metabarcoding derives from Shelton et al. [35], building on the work of others [19, 25, 33, 34, 37]. Briefly, we envision a metabarcoding dataset as compositional, arising from a chain of sampling and amplification processes acting on individual DNA molecules. In the present model, variance in read abundance among technical replicates of

metabarcoding data is a function of (1) stochastic sampling of rare molecules from the DNA extraction prior to PCR, (2) deterministic amplification biases during PCR, and (3) stochastic sampling of amplicons during sequencing. We note that Shelton et al. [35] only modeled processes (2) and (3) and did not account for subsampling from the DNA extraction into the PCR reaction (1).

We start with a sample of extracted DNA containing sequences from multiple species. From this starting point, there are many different metabarcoding laboratory protocols that lead to observed sequences from a sequencing instrument [41, 42]. Here, we develop a model using three main stochastic processes following the commonly used two-step PCR library generation process (e.g., a target PCR followed by an indexing PCR). First, we assume a sample of DNA is extracted and included in a multi-taxon PCR reaction. Second, PCR amplification using a specific primer and protocol occurs, replicating the DNA molecules for each taxon. This second step includes the various target PCR, cleaning, indexing PCR, and pooling steps that occur during or following the main PCR reaction. Finally, the resulting mix of DNA amplicons is sampled to generate a compositional sample of amplicons that are observed after sequencing.

Mathematically, we can write a simulation for this framework as a series of linked stochastic processes. Suppose there is a solution of DNA molecules of concentration $\lambda_i$ copies per μL from the $i$th taxon, $i = 1,2,\ldots,I$. Let $W_{ij}$ be the discrete number of DNA molecules for species $i$ sampled (i.e., in the tube in which a given PCR reaction takes place) of technical replicate $j$ at the beginning of PCR and

$$W_{ij} \sim Poisson(\lambda_i V) \tag{1}$$

Here, $V$ is the volume ($\mu L$) of template DNA sampled from the DNA extract. This equation assumes each taxon is sampled independently. Note that different technical replicates (e.g., $j = 1$ and $j = 2$) arise from the same environmental sample but may contain different numbers of molecules for a given species due to sampling variability. Importantly, we note that the Shelton et al. [35] mechanistic framework did not incorporate this stochastic subsampling component in the underlying model, only focusing on the deterministic PCR reaction described below.

Next, we model a three-step PCR process assuming a two-step PCR process with a sub-sampling and PCR cleaning process in between. Most importantly, we assume the amplicons produced during a PCR reaction are influenced by a species-specific amplification efficiency $a_i$, which is characteristic of the interaction between the particular primer set used, reaction chemistry, and template molecule of each species ($i$) being amplified [35]. For any species, $X_{1ij}$ is the expected number of amplicons present in a technical replicate at the end of the first PCR. $X_{1ij}$ is directly related to the efficiency of amplification and the starting number of DNA molecules, $W_{ij}(1 + a_i)^{N_{pcr}}$, where $N_{pcr}$ is the number of PCR cycles and $a_i$ is bounded on (0,1); $a_i = 1$ represents a perfect doubling of molecules with each PCR cycle. For the purposes of this paper, $X_{ij}$ can be modeled at each step as:

$$X_{1ij} \sim Poisson\left(W_{ij}(1 + a_i)^{N_{pcr1}}\right) \tag{2}$$

$$X_{2ij} \sim Binomial\left(\pi, X_{1ij}\right) \tag{3}$$

$$X_{3ij} \sim Poisson(X_{2ij}(1 + 0.9)^{N_{pcr2}}) \tag{4}$$

where $\pi$ is the proportion of the first PCR product used in the second PCR amplification. The

above Eq 2 is a Neyman Type A equation also known as a Poisson-stopped sum distribution which has been frequently used to model overdistributed and zero-inflated biological data sets [43–46]. Such approaches have been applied to described overdispersed molecular datasets [47–51], although we are unaware of any direct applications to PCR amplification specifically. Note that during the indexing reaction (Eq 4) all species share a single amplification efficiency ($a_i = 0.9$) as we assume all indexing primers anneal to the indexing adapter sequences with equal efficiency. Here $X_3$ is the number of amplicons present after both PCR amplifications but before sequencing. Finally, the sequencing instrument generates a total number of reads within technical replicate $j$ ($N_{readsj}$) and each replicate has a vector of observed read counts ($Y_j$, bolding indicates vectors) for $I$ species.

$$Y_j \sim Multinomial\left(p_j,\ N_{reads,j}\right) \tag{5}$$

where $p$ is the proportion of reads from species $i$ in technical replicate $j$, and $p_{ij} = \frac{X_{3ij}}{\sum_{i=1}^{I} X_{3ij}}$.

Thus, observed read counts ($Y_{ij}$) are sampled stochastically based on their relative amplicon abundances, $p_{ij}$.

The above model provides a general framework for understanding the causes of variability in observed read counts for a given species in a set of technical replicates ($Y_{ij}$). Specifically, the probability of non-detections (i.e., $Y_{ij}$ is 0 for some $j$ replicates and non-zero for others) can be attributed to 1) the initial DNA concentration $\lambda_i$ and 2) the amount of species-specific variation in amplification efficiency ($a_i$). The simulation allows us to identify two distinct causes of non-detection, $p(Y_{ij} = 0)$ when $Y_{ij} > 0$ for a sister technical replicate. First, non-detection may occur through the initial subsampling process from the DNA extraction where no molecules of species $i$ are included in the initial PCR ($W_{ij} = 0$, in which case we are interested in $p(W_{ij} = 0)$ because $p(Y_{ij} = 0|W_{ij} = 0) = 1$). Second, zeros can arise due to PCR amplification and sequence-sampling processes where molecules from species $i$ are present in the DNA extraction but are not successfully amplified and sequenced; thus we are interested in $p(Y_{ij} = 0|W_{ij} > 0)$. While $p(W_{ij} = 0)$ is trivial to calculate from Eq 1, determining $p(Y_{ij} = 0)$ and $p(Y_{ij} = 0|W_{ij} > 0)$ is not. We turn to simulations to understand the contributions of variation in $\lambda_i$ and $a_i$ to the probability of non-detection.

We simulate four communities with different levels of species richness (N = 4, 10, 30, and 50 species). For simplicity, we assume all taxa start with identical DNA concentrations regardless of the richness. DNA concentrations, $\lambda_i$ vary from 0.5 to 10,000 copies $\mu L^{-1}$ (for simplicity we set the volume of DNA extract used for PCR $V = 1$ uL for all simulations). We further allow a range of amplification efficiencies ($a_i$) among taxa where $a \sim Beta(0.7\gamma, 0.3\gamma)$ with $\gamma$ ranging from 5 (high variation among species) to 1x 10^6 (no variation among species), but with a constant average amplification efficiency of 0.7 for all scenarios. We simulated 50,000 realizations for each combination of richness (4 levels), $\lambda$ (18 levels), and $\gamma$ (6 levels: 5, 10, 20, 100, 100, 1 million)), for a total of 432 scenarios. For all the simulations, we allowed sequencing depth to vary among replicates $N_{read,j}$ was uniformly drawn from discrete values between 60,000 and 140,000), used a fixed sampling fraction ($\pi = 0.20$), and fixed number of PCR cycles ($N_{pcr1} = 35$ and $N_{pcr2} = 10$). We calculated a range of summary statistics for each scenario, including the overall probability of non-detection, $p(Y_{ij} = 0)$; the probability of non-detection due to the absence of the target molecule, $p(W_{ij} = 0)$; and summaries of read counts both in absolute terms and in terms of relative abundance. See S1 File for simulations exploring the effects of sampling depth on non-detections and S2 File for simulations using alternative model parameters changing $N_{pcr1}$ and uneven DNA concentrations.

We reiterate that this manuscript builds off the work of Shelton et al. [35] and Gold et al. [39] to use the above expanded mechanistic framework to explore the effects of both subsampling and PCR processes on the patterns of non-detections. Shelton et al. [35] defined the underlying mechanistic framework for metabarcoding, demonstrating the importance of accounting for amplification efficiencies in general as well as presenting a series of case studies using mock communities to validate the underlying mechanistic framework. Gold et al. [39] demonstrated that quantitative abundance estimates can also be estimated using a joint model of both metabarcoding sequences and morphological counts of ichthyoplankton using independent abundance estimates to constrain possible amplification efficiency parameters. Both works highlight the value of accounting for amplification biases in deriving quantitative abundance estimates, but neither of these works model or account for the subsampling of rare DNA molecules from the DNA extraction prior to PCR amplification in addition to the PCR and sequencing processes. Here, we build upon these works by directly modeling the subsampling process prior to PCR amplification, directly testing the effects of subsampling and PCR processes on the patterns of non-detections.

## Empirical testing

We explore the effects of subsampling and PCR processes on non-detections in metabarcoding data making use of two empirical examples: mock community and CalCOFI. Each empirical example has three data streams generated from a common set of biological samples: input DNA molecules, metabarcoding data, and amplification efficiency estimates for the relevant species. The amplification efficiency estimates used in both examples are derived from the mock community dataset. For the mock community example input DNA molecules are known, but for the CalCOFI example, we use organismal abundance (as a proxy for input DNA molecules). We obtain these underlying data streams from Shelton et al. [35] and Gold et al. [39] as well as additional mock community sequencing data uniquely presented here. No permits were needed for this work as we are utilizing previously published data sets.

## CalCOFI example study design

As part of the California Cooperative Oceanic Fisheries Investigations (CalCOFI), Gold et al. [39] use morphological and molecular methods to analyze the response of ichthyoplankton in the California Current Large Marine Ecosystem to ocean warming. Ichthyoplankton samples were collected in oblique bongo net tows on CalCOFI research cruises over two decades (1996; 1998–2019). We note that bongo nets have two paired 505 μm nets that are concurrently sampling immediately adjacent water columns. Once a sampling tow concluded, all zooplankton contents present on one side of the bongo net were preserved in Tris-buffered 95% ethanol for metabarcoding-derived species identification and sequencing. Contents on the other side of the paired net were preserved in sodium borate-buffered 2% formaldehyde for microscopy-derived species identification and abundance (number of larvae per species per jar). This dataset yields paired samples for both metabarcoding analysis and absolute abundance counts from the same sampling event.

## Abundance estimation via microscopy from CalCOFI samples

Formalin-preserved larvae were identified and enumerated from 84 formalin preserved samples following the methods of Thompson et al. [52]. The majority of taxa (83%, 76/92) were identified to species level. Here we assume that the relationship of absolute abundance (counts of individual species) is proportional to the amount of species-specific DNA in the extraction. See the discussion for the merits of this assumption.

## Metabarcoding data generation from CalCOFI samples

Ethanol preserved samples were stored in the Pelagic Invertebrate Collection at Scripps Institution of Oceanography for metabarcoding analyses [53]. DNA sequences were generated from the liquid ethanol pipetted off the top of 84 preserved samples as described in Gold et al. [39]. Briefly, up to 125 mL of ethanol (mean = 121 mL, n = 6 < 125 mL, min = 34 mL) was filtered onto 0.2 μm PVDF filters and were extracted using a Qiagen DNeasy Blood and Tissue kit. We then amplified three technical PCR replicates using a touchdown PCR and the MiFish Universal Teleost specific primer [54]. Both a negative control (molecular grade water instead of DNA extract) and two positive controls (DNA extract from non-native, non-target species) were included alongside samples. Libraries were prepared using Illumina Nextera indices following the methods of Curd et al. [55] and sequenced on a NextSeq 2x 150 bp mid output. Sequencing data was then processed using the *Anacapa Toolkit* [55] to conduct quality control, ASV dereplication, and taxonomic assignment. Sequences were annotated with the California fish specific reference database and a bootstrap confidence cutoff score of 60 following the methods of Gold et al. [56]. Eight technical replicates with either low sequencing depth (n<30,000) or high dissimilarity (Bray Curtis dissimilarity > 0.7) were removed.

## Amplification efficiency estimation from mock communities

We used a subset of the mock communities generated for Shelton et al. [35] alongside two additional mock communities presented here to estimate amplification efficiencies of relevant fish species. Mock communities included DNA from 57 voucher fish tissue samples, 17 of which were detected in the CalCOFI metabarcoding data set, from the Scripps Institution of Oceanography Marine Vertebrate Collection. To accurately quantify input DNA for each species within the mock community, we used a nested PCR strategy in which mock communities were generated by pooling resultant longer fragment PCR products of each species rather than by pooling the total genomic DNA of each species (which includes variable amounts of nDNA as well as bacterial and other DNA sources). To implement our nested PCR strategy, we first amplified a 612 bp fragment of the *12S* rRNA gene that contains the MiFish Universal Teleost *12S* primer set [57], and quantified the resulting PCR products using the QuBit Broad Range dsDNA assay (Thermofisher Scientific, Inc.); this yielded measurements of species-specific, amplifiable DNA. Using this known-concentration DNA we generated 9 distinct mock communities by pooling long fragment PCR products comprising three distinct sets of species and three abundance distributions (See S1 Table in S3 File). Pooled mock communities were quantified using the QuBit Broad Range dsDNA assay (estimated concentrations ranged from 8–12 ng μL$^{-1}$) and then diluted serially by a 1:10 dilution down to $10^{-8}$ original concentration. The two additional mock communities included in this study were derived from the same "coastal even" mock community pool but diluted serially by a 1:10 dilution down to $10^{-4}$ and $10^{-5}$ original concentration to explore the subsampling and PCR process effects in higher DNA concentrations. We then converted ng μL$^{-1}$ to copies μL$^{-1}$ using the following equation:

$$\text{Copies } \mu L^{-1} = \frac{\text{QuBit Concentration [ng } \mu L^{-1}]*6.022\text{x}10^{23} \text{ [molecules mol}^{-1}]}{612 \text{ [bp]}*650 \text{ [g mol}^{-1} \text{ bp}^{-1}]*(1\text{x}10^{9} \text{ [ng g}^{-1}])}$$

Finally, input concentrations of 380–4,268,989 DNA copies μL$^{-1}$ for each community were used as template in the MiFish Universal Teleost *12S* PCR step, targeting a ~185 bp fragment within the larger 612 bp PCR fragment (S2 Table in S3 File). We amplified each of the mock communities in triplicate following the methods of Curd et al. [58] and each triplicate PCR technical replicate was sequenced separately. Metabarcoding libraries were then prepared and

sequenced on a MiSeq platform using a v3 600 cartridge following the methods of Gold et al. [59] across two sequencing runs. Resulting sequences were processed using the *Anacapa Toolkit* using the global *CRUX* generated reference database given the broad geographic distribution of species from Gold et al. [56]. We also used the same taxonomic cutoff score of 60 used for the larval metabarcoding data. Taxonomic assignment of ASVs was confirmed with BLAST using default settings. For the two observed discrepancies between *Anacapa classifier* and BLAST annotations, we chose to use BLAST assignments with greater than 99% identity and 100% query length match as they matched our known vouchered specimen identifications.

We fit the model from Shelton et al. [35] to a third of the data (3 technical replicates of each evenly pooled mock community) to calculate amplification efficiencies for each species in the mock communities. Generated parameter estimates were then used to predict the starting proportions of DNA in the remaining two-thirds of the data, for an out-of-sample estimate of accuracy. We used the resulting model output to calculate the mean amplification efficiency per species. The model, implementation, and code are detailed in Shelton et al. [35], but there are two particularly relevant points from the model for connecting the simulation and empirical results that we highlight here. While we simulate absolute amplification efficiencies ($a_i$), because metabarcoding data is compositional, the absolute amplification efficiency cannot be estimated from metabarcoding data. Instead, we estimate amplification efficiencies for each species relative to a reference efficiency (see also [33, 34]). In our case we estimate $\alpha_i$ as the amplification efficiency of species $i$, $a_i$, relative to the efficiency of a reference species, $a_R$, therefore $\alpha_i = \frac{a_i}{a_R}$. Thus, for simulations we discuss $a$ but for empirical data, we discuss $\alpha$. Note while values of $\alpha$ can be directly calculated from $a$, values of $a$ are not uniquely identifiable from $\alpha$. We note that these amplification efficiency estimates derived from the mock communities are used in both the mock community example as well as the CalCOFI example.

All data and code for conducting analyses are publicly available via NCBI SRA (BioProject: PRJNA966238), Dryad (https://doi.org/10.5068/D1XH5H), and GitHub (https://doi.org/10.5281/zenodo.7887383).

## Hypothesis testing

We explore the effects of both subsampling and PCR processes on non-detections and sequence patterns. We hypothesize that non-detections can arise from both processes, but that the dominant process depends on the rarity of the molecule. For rare DNA molecules, subsampling will drive the occurrence of non-detections, while for abundant molecules, poor amplification efficiencies during PCR will drive the occurrence of non-detections. We test these hypotheses using simulation and the mock community and CalCOFI empirical examples described above.

# Results

## Simulation results

We found a strong correlation between the probability of non-detections and both the absolute abundance of template DNA molecules and amplification efficiencies (Fig 2). The probability of non-detections ($p(Y = 0)$) dramatically declines when concentrations of template DNA are greater than ~10 copies $\mu L^{-1}$ per species, given an average amplification efficiency of 0.7 (Fig 2C & 2D). Likewise, our results demonstrate that species with low amplification efficiencies exhibit high probabilities of non-detections regardless of starting DNA concentrations (Fig 2A and 2B). Importantly, we demonstrate that even species with an amplification efficiency

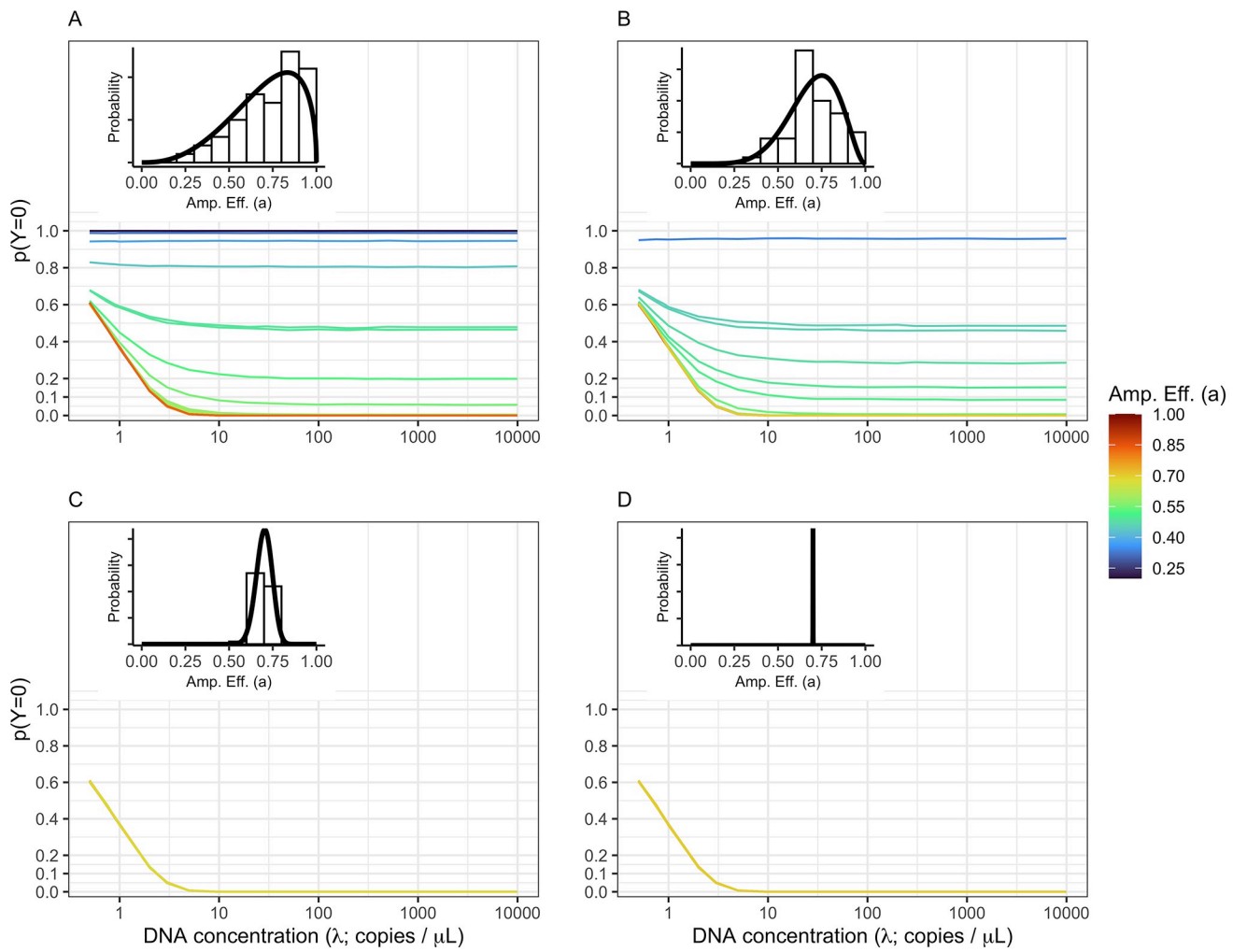

**Fig 2. Non-detections driven by both DNA concentration and amplification efficiency.** The probability of non-detection ($p(Y = 0)$) is shown for a community of 50 equally abundant taxa with four different distributions of amplification efficiency across taxa. The amplification efficiency (Amp. Eff. (a)) distribution of each example is inset in the upper left of each panel. The amount of among-taxa variation in amplification efficiency varies from high variation ($A$; $\gamma = 5$) to moderate variation ($B$: $\gamma = 10$) to low variation ($C$: $\gamma = 100$) to effectively no variation ($D$: $\gamma = 1,000,000$). Both subsampling and amplification efficiencies influence the rate of non-detection. The probability of observing no DNA in a given technical replicate is highest at low DNA concentrations ($<10$ copies /μL). However, non-detections are possible for species with below average amplification efficiencies (in this case approximately $a_i = 0.7$) and very likely ($p(Y = 0) > 0.5$) for amplification well below average ($a_i < 0.4$).

slightly below average (i.e., $a < 0.7$) exhibit high rates of non-detections at DNA concentrations far higher than from typical eDNA field samples (e.g., $\lambda > 100$ copies/μL; [58]). The simulations indicate that the probability of non-detection is dominated by the subsampling process at low template DNA concentrations, while the probability of non-detection is driven primarily by the PCR process (i.e., differences in amplification efficiencies) at higher template DNA concentrations.

## Empirical results

**Mock community example results.** The mock community data set consisted of 1.3 million amplicon sequence reads across two sequencing runs that passed through the *Anacapa Toolkit* quality control, ASV dereplication, and decontamination processes. A total of 33

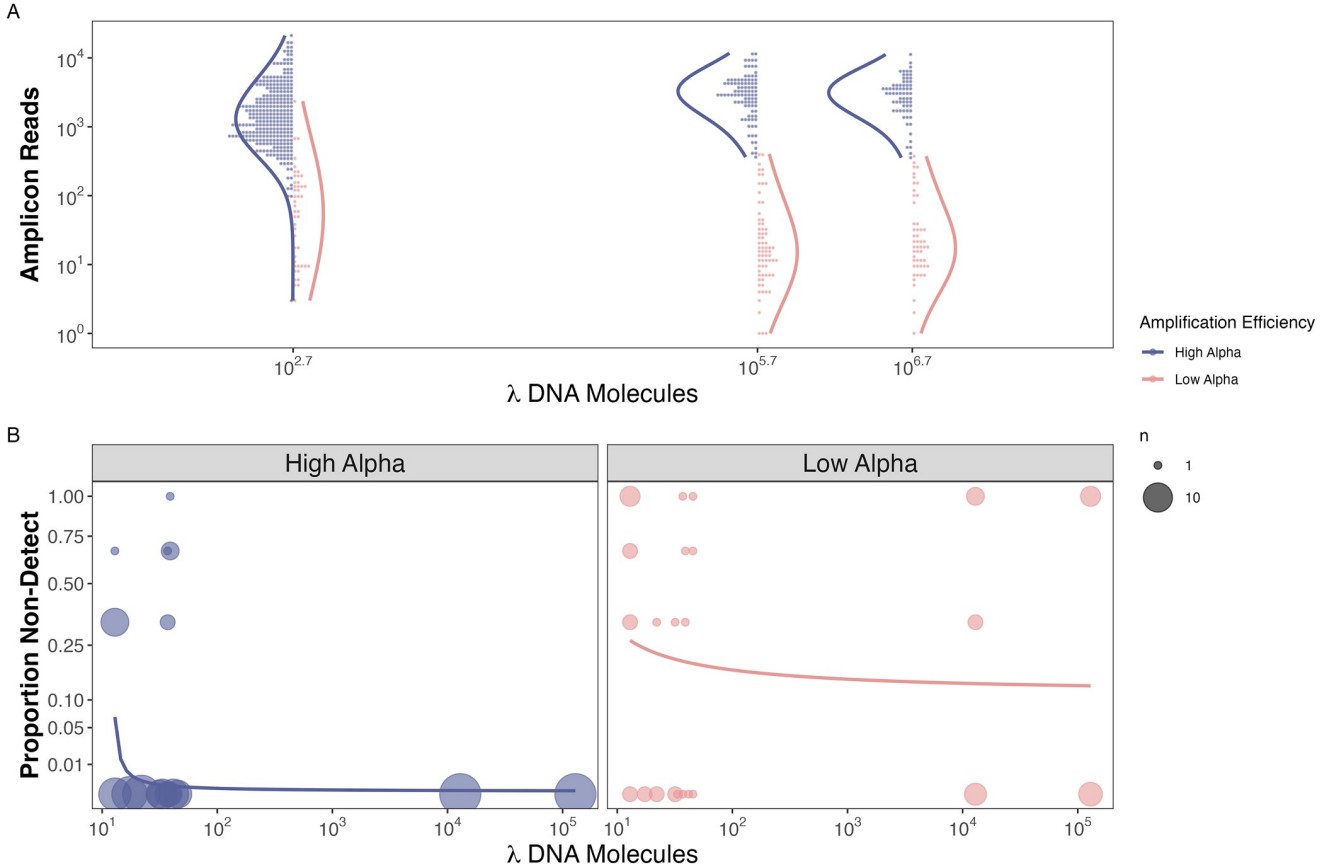

**Fig 3. Observed reads and non-detections are a function of amplification efficiency and input DNA concentration in the mock community example.** For species observed within a replicate, we find that species with higher amplification efficiencies ($\alpha_i > 0.7$) have a greater number of observed reads for an equivalent template DNA concentration (Panel A). We also find no difference in the total number of observed reads and increased DNA concentration, as expected for a compositional data set. Furthermore, we find a greater proportion of non-detections when both DNA concentration and amplification efficiencies are lower (Panel B).

unique samples comprising three distinct community assemblages each with three PCR technical replicates were sequenced. Sequencing depth ranged from 9,872 reads to 98,744 reads per technical replicate. Of the 57 voucher species represented, we classified 56 unique species, and used the Shelton et al. [35] model to estimate amplification efficiencies for each species. One species, *Urobatis halleri*, was not detected in any technical replicate. *Citharichthys sordidus* was present in all mock communities and was selected as the reference species for estimating relative amplification efficiencies. Across all species, $\alpha_i$ ranged from -0.30 to 0.03 with a mean of -0.06 (S3 Table in S3 File). For presentation purposes, we label species with $\alpha_i$ values below -0.07 as a low amplification efficiency group (n = 15) and the remaining species as a high amplification (n = 41).

**Mock community hypothesis testing.** From our mock community results, we found that the probability of non-detections is strongly correlated with both the abundance of DNA molecules for a given species within a sample and the species-specific amplification efficiency (Fig 3). Non-detections occur more frequently at low DNA concentrations regardless of amplification efficiency (Fig 3B). Species with lower amplification efficiencies ($\alpha_i < -0.07$) had higher rates of non-detections even at high input DNA concentrations ($10^4$ copies $\mu L^{-1}$) than species with higher amplification efficiencies ($\alpha_i > -0.07$) (Fig 3B). Furthermore, species with higher

amplification efficiencies ($\alpha_i > -0.07$) have higher observed sequence read counts for an equivalent template DNA concentration, whether that template concentration is high or low (Fig 3A). The 41 species with high amplification efficiencies have more reads sequenced per DNA molecule added (mean ± sd = 4.1 ± 6.31, range = 0.00–55.4) than the 15 species with low amplification efficiencies (mean ± sd = 0.1 ± 0.47, range = 0.00–5.5).

**CalCOFI example results.** Independent estimates of species abundance were generated from sorting 9,610 larvae from 84 jars (min = 2, max = 960). See Gold et al. [39] for a detailed description of the microscopy results.

The metabarcoding data set generated from ethanol-derived eDNA consisted of a total of 54.5 million amplicon sequence reads that passed through the *Anacapa Toolkit* quality control, ASV dereplication, and decontamination processes. Sequencing depth ranged from 36,050 reads to 1.2 million reads per technical replicate. For our integrated Bayesian model of the probability of non-detection in a technical replicate, we focused on 17 species that had 1) sufficient representation across the metabarcoding data set (observed in > 10 technical PCR replicates) to achieve model convergence and 2) were represented in our mock communities (see below). See Gold et al. [39] for the full description of model implementation and results.

**CalCOFI hypothesis testing.** Within the CalCOFI empirical example, we found that the probability of non-detections is strongly correlated with both larval counts (assumed proportional to the abundance of DNA molecules) for a given species within a sample and the species-specific amplification efficiency (Fig 4B). We could not explore the relationship between non-detections and amplification efficiency in the CalCOFI example as no species exhibiting lower amplification efficiencies ($\alpha_i < -0.07$, n = 2) were observed in high abundance (max = 9 larvae in a jar; Fig 4B). However, we did observe that species with higher larval counts in ethanol-preserved samples from plankton tows also have higher observed sequence read counts (Fig 4A). Similar to the mock community results, the 15 species with high amplification efficiencies have more reads sequenced per larvae counted (mean ± sd = 6,689 ± 28,305, range = 0–79,454) than the two species in the low amplification efficiency group (mean ± sd = 524 ± 1,080, range = 0–7,101).

## Discussion

Using both simulated and empirical data, we demonstrate that observed sequence read counts from metabarcoding data are a function of input DNA concentrations, subsampling, and species-specific amplification efficiencies. Variability among replicates in detections of specific taxa–reflecting either rare targets or poor amplification efficiencies–are a substantial source of noise in these data. Consequently, it can be difficult to distinguish signal from noise in metabarcoding datasets. Our results illustrate several potential causes of non-detections and suggest that metabarcoding data can provide reliable quantitative estimates for species with abundant input DNA (> ~50 copies $\mu L^{-1}$) and high species-specific amplification efficiencies. By characterizing underlying sources of sequence read count variability in metabarcoding, we identify key sources of noise that impact our ability to derive quantitative estimates of starting DNA concentrations.

### Subsampling rare targets results in non-detections

Consistent with expectation, our framework strongly suggests that, all else being equal in a metabarcoding assay (e.g., assuming even amplification efficiencies across species), rarer template DNA molecules have a higher probability of non-detection across technical replicates. These findings align well with observations of qPCR assays in which the probability of non-

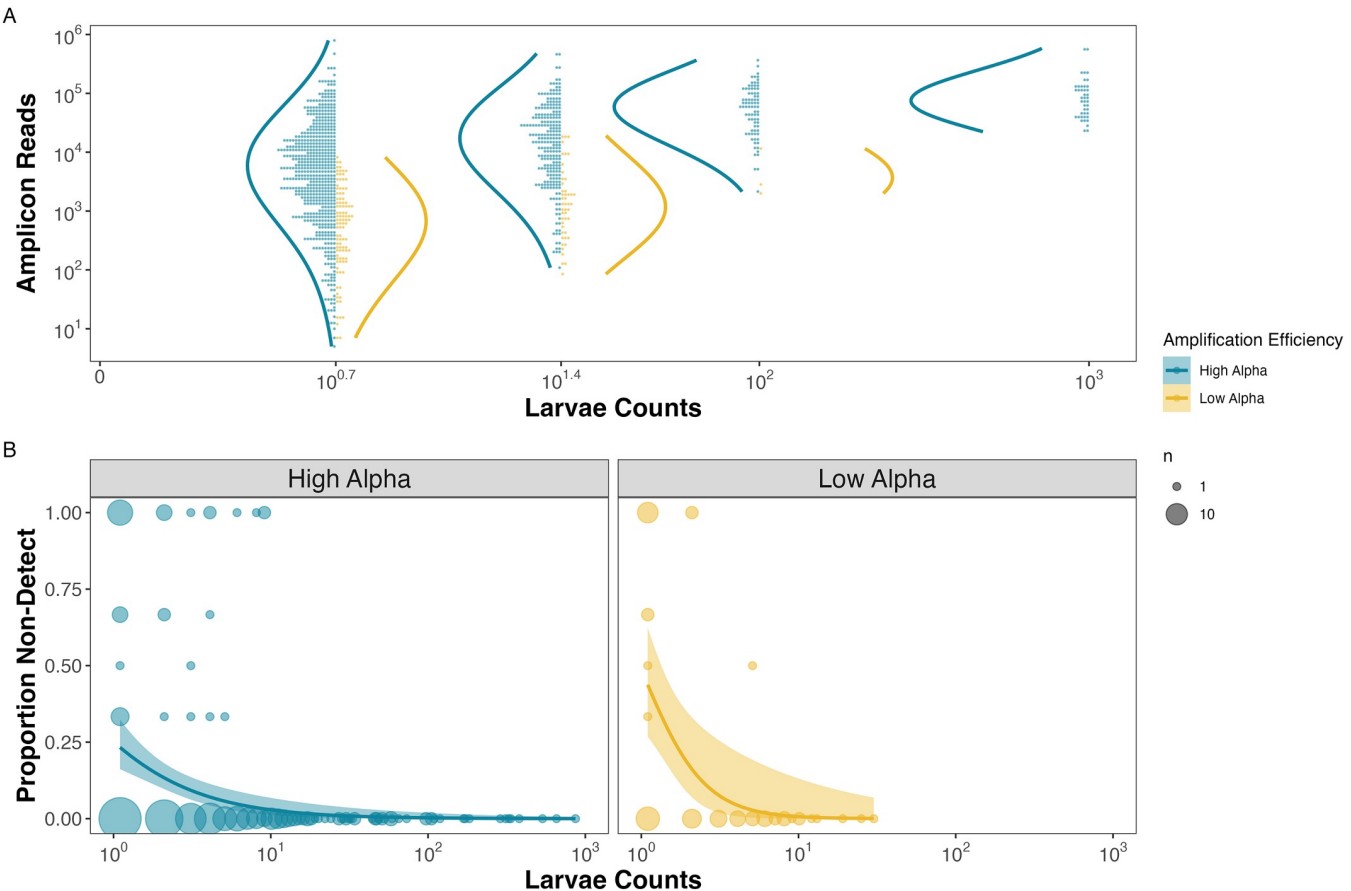

**Fig 4. Observed reads and non-detections are a function of amplification efficiency and larval abundances in the CalCOFI example.** For species observed within a replicate, we find that species with higher amplification efficiencies ($\alpha_i > $-0.07) have consistently greater numbers of observed reads for an equivalent template DNA concentration (Panel A). We assume that the number of larvae in the jar is proportional to the number of DNA molecules present. We also find a greater proportion of non-detections when larvae are rare in the jars (Panel B).

detection increases as you approach the limit of detection, in terms of absolute copies of DNA per reaction volume [59, 60]. High rates of non-detections in qPCR assays are commonly observed for input DNA concentrations between 1 and 10 copies [59, 61, 62] and are likely driven by subsampling errors in which too few or no physical DNA molecules are transferred into a given PCR reaction [63–65]. The findings from both simulated and empirical metabarcoding results reported here reflect a similar pattern.

Importantly, subsampling rare DNA molecules yields stark differences in observed read counts among technical replicates, non-detections being the most obvious case of this phenomenon [23, 24, 40, 59]. Together, these findings strongly support the hypothesis that the concentration of target DNA within a sample influences the observed patterns of amplicon read counts, particularly increasing the probability of non-detections for species with low template DNA concentrations. While the relationship between template concentration and non-detections is well documented in the qPCR literature [18, 20, 24, 32, 40, 59], such observations of high rates of non-detections due to subsampling processes prior to PCR amplification are important for the field of metabarcoding because they strongly justify the use of overdispersed multinomial sampling approaches within future mechanistic models [35].

## Amplification efficiencies drive sequence counts and non-detections

In contrast to non-detections of rare DNA molecules, the causes of non-detection among species with abundant template concentrations are not widely appreciated. Both our simulation and empirical mock community results demonstrate a clear association between the probability of non-detections and amplification efficiencies, where species with higher amplification efficiencies exhibit fewer non-detections. Unfortunately, we could not test these patterns in the empirical CalCOFI results because we only observed low amplification efficiencies in two species and neither of these species were present in high abundance in any sampled jars (Fig 4). However, the strength and similarity in the patterns in both the simulation and empirical mock community results strongly suggest that the PCR process drives non-detections for species with abundant DNA.

Furthermore, our simulation and empirical results demonstrate that species with higher amplification efficiencies have higher observed amplicon read counts, confirming the predictions of previous compositional modeling efforts [33–35]. This was best exemplified in the mock community dataset where species with higher amplification efficiencies had over an order of magnitude more sequence reads for a given concentration of DNA than species with lower amplification efficiencies. Although only two of the 17 species in the CalCOFI data had low amplification efficiencies, the fact that we also observed an order of magnitude difference in average sequence reads per larvae between species with higher and lower amplification efficiencies strongly supports our conclusion that amplicon reads counts are a function of amplification efficiencies in addition to starting concentration.

The source of this observed variation in amplification efficiency among species in metabarcoding approaches arises from complex PCR processes, including primer specificity, DNA polymerase selectivity, annealing temperature, GC content, and higher-order dimensional structure of DNA, inhibition, and co-factors such as $MgCl_2$, among others [66–72]. However, the relative importance of each of these processes on amplification biases and whether amplification biases can be mitigated remains unknown. At present, this complexity makes designing metabarcoding assays that are highly specific for only target taxa challenging [73, 74], resulting in the amplification of off-target taxa as well as a range of amplification efficiencies across target taxa [34, 35, 56, 75, 76]. Our simulations and empirical results demonstrate that a range of amplification efficiencies can result in substantial noise in metabarcoding data sets and that accounting for such amplification bias is important for the accurate interpretation of metabarcoding results.

## Accurately accounting for non-detections in metabarcoding data

Our results illustrate that noise in metabarcoding datasets, like signal, is non-random and can be accounted for [24], but that alone, metabarcoding data is insufficient to tease apart these complex interactions. Here we demonstrate that distinguishing signal from noise in metabarcoding datasets is tractable using independent estimates of amplification efficiencies and underlying DNA concentrations. Amplification efficiencies can be estimated by either generating mock communities [34, 35], by amplifying a subset of samples multiple times at various numbers of PCR cycles [33], or by including internal positive controls within each PCR [36]. Likewise, underlying DNA concentrations can be estimated using qPCR or dPCR assays of key taxa or the metabarcoding locus itself [77]; or estimated using non-genetic independent abundance estimates such as the microscopy counts presented in Gold et al. [39]. As demonstrated here, and in Shelton et al. [35], McLaren et al. [34], and Silverman et al. [33], the inclusion of independent estimates of amplification efficiencies and DNA concentrations allow for the delineation of signal from noise from metabarcoding data sets. Further modeling efforts

incorporating stochastic sampling of rare molecules prior to PCR will allow for accurate quantification and identification of true absences in metabarcoding data sets, greatly enhancing biological and ecological interpretation.

Furthermore, our analysis also underscores the importance of technical PCR replicates to quantify variability in sequence read counts in metabarcoding studies [78–80]. Without technical replicates, we would not have been able to quantify the frequency of non-detections in our metabarcoding datasets [25]. Current best practices for qPCR and dPCR assays include numerous technical replicates to help distinguish signal from noise [59, 61]. However, we recognize that technical replication dramatically increases the cost and effort of metabarcoding projects and may exhaust limited DNA extracts and resources. Alternatively, technical replicates could be performed on a subset of samples and the observed variance could be used to contextualize sequence read patterns in the whole dataset. However, such approaches come with a suite of assumptions, particularly whether the pattern of species' sequence counts behaves similarly across all samples and environments/treatments. Future efforts to validate such approaches are clearly warranted.

In addition, given the importance of subsampling in driving non-detections, our results strongly suggest that field and laboratory processes that increase the absolute abundance of DNA molecules will reduce the noise in observed amplicon sequence reads [81]. For example, using a greater volume of DNA template for PCR reactions (3 μL vs. 1 μL) will reduce subsampling driven non-detections across samples. Likewise, increasing the total amount of water filtered for eDNA samples (3 L vs. 1 L) acts to concentrate DNA from the environment, similarly reducing subsampling driven non-detections [82]. These are two of many examples of laboratory protocols that may serve to increase the available number of DNA molecules and reduce the impacts of subsampling rare molecules, consequently improving quantitative estimates from amplicon sequence data.

The above mechanistic frameworks focus on processes from DNA extraction through sequencing, but do not approach the myriad of factors that influence the amount of DNA collected from the environment, gut, or other starting communities for metabarcoding (Fig 1). Substantial efforts have focused on understanding the effects of gene copy number, patchiness, shedding and degradation rates, and the fate and transport of cellular DNA, among others, on the amount/types of DNA collected from the environment [34, 65, 83]. Furthermore, limitations in bioinformatics approaches can also compound these issues, particularly in determining which sequences are retained and analyzed [56, 84–87]. Linking such research to the growing body of work that quantifies sources of potential bias in the lab, including the present study, is an important next step in understanding the relationship between biological signals and observed sequence read counts.

We recognize that incorporating the additional laboratory analyses and technical replicates to better characterize metabarcoding results may not be feasible for all metabarcoding applications. Many metabarcoding efforts are exploratory in nature, primarily focused on the characterization of biodiversity in under sampled habitats including the deep sea, polar regions, remote alpine regions, etc. For such exploratory biodiversity surveys, the additional efforts needed to achieve quantitative metabarcoding outlined above may not be practicable given surveying and budget constraints. However, it is important to recognize that our framework extends not only to quantitative metabarcoding but detection rates of taxa from metabarcoding surveys. The expected detection rate (observed reads > 0) of a given species in metabarcoding data is a function of the following parameters: the other species in the community, the amplification rate of the target species, the amplification rates of other species, the proportional abundance of the target species, and the absolute abundance of the target species as demonstrated in our empirical datasets above. Thus, estimating the probability of detection

from metabarcoding data alone is difficult in the abstract, but is quite tractable given a set of estimated parameters for a particular sampled community (e.g. estimates of species specific amplification efficiencies and input template DNA concentrations). Conversely, this presents a challenge for exploratory metabarcoding applications within systems with limited ecological context as species detection rates are a function of multiple unsampled parameters. Thus researchers should employ judicious caution in the interpretation of uncorrected metabarcoding sequence read counts in such exploratory analyses [35, 37, 84].

Undoubtedly, addressing this shortcoming of compositional metabarcoding data requires increased field and laboratory efforts. Such challenges are acute in under studied systems where the creation of mock communities is particularly difficult with limited access to vouchered DNA samples, let alone known species lists. However, exploratory metabarcoding studies do not preclude the revisiting of quantitative metabarcoding approaches in the future, especially since DNA extracts can be archived. For example, metabarcoding data can be generated first to provide an initial perspective into community assemblages that then allows for the identification and development of single species qPCR/dPCR assays and mock communities or variable PCR targets. In summary, we argue that all future best practices of metabarcoding results incorporate additional independent estimates of amplification efficiency, independent estimates of DNA concentrations, and technical replicates to better contextualize metabarcoding efforts. Given the rapid decline in sequencing costs and steady improvement in the development and implementation of molecular assays, such additional work is tractable, opening the door to adoption for routine application across metabarcoding studies to generate characterization of underlying biological communities.

## Conclusion

Ultimately, we demonstrate that variation in amplification efficiencies and underlying template DNA concentration are responsible for a substantial portion of observed noise in metabarcoding datasets. This study demonstrates the value of incorporating additional independent estimates of amplification efficiencies and DNA concentration along with amplicon sequence data, providing for the application of routine statistical approaches and straightforward interpretation of observed read patterns. Together with Shelton et al. [35], we provide a framework for establishing reliable estimates of abundance from amplicon sequence data that will be critical for extending the application of this method to health and ecological questions.

## Supporting information

**S1 File. A note on sampling depth and the probability of observing zeros.**
(DOCX)

**S2 File. Alternate simulation results.** We changed parameters in the simulation to understand their effects on non-detection. Specifically, we tested the effect of $N_{pcr1} = 20$ rather than $N_{pcr1} = 35$ presented in the main text. We also simulated uneven DNA concentrations across species and its effect on the probability of non-detection.
(DOCX)

**S3 File. File consists of all supporting tables.** S1 Table contains the input concentration of DNA into each mock community. S2 Table contains the calculations used to derive copies per species for each mock community based on the Qubit Concentration. S3 Table contains the summary of amplification efficiency model estimates.
(XLSX)

**S1 Fig. Non-detects driven by both DNA concentration and amplification efficiency.** The probability of non-detection ($p(Y = 0)$) is shown for a community of 50, equally abundant taxa with the amplification efficiency distribution shown inset in each panel. This simulation uses $N_{pcr1} = 20$ (see Fig 1 for the same simulation but with $N_{pcr1} = 35$). The amount of among-taxa variation in amplification efficiency varies from highly variable (*A*; γ = 5) to moderate variation (*B*: γ = 10) to low variation (*C*: γ = 100) to effectively no variation (*D*: γ = 1,000,000). Both subsampling and amplification efficiencies influence the rate of non-detection. The probability of observing no DNA in a given technical replicate is highest at low DNA concentrations (<10 copies /μL). However, non-detects are possible for species with low amplification efficiencies and very likely ($p (Y = 0) > 0.5$) for amplification well below average (in this case approximately $a_i < 0.3$). (TIFF)

**S2 Fig. Non-detects driven by both DNA concentration and amplification efficiency.** The probability of non-detection ($p(Y = 0)$) is shown for a community of 20 taxa with 4 taxa comprising 0.20 of the initial DNA, 8 with 0.05 of the DNA, and 10 species comprising 2% of the DNA across a range of initial DNA concentrations. *A*: Presents results for a single 20 taxa community with facets representing the three abundance categories. *B* shows results for 20 communities of 20 taxa each to illustrate general patterns. For all simulations we use $N_{pcr1} = 35$ and a fixed amount of among-taxa variation in amplification efficiency (γ = 5). Clearly, relative abundance influence the rate of non-detection with relatively rare taxa (those with 0.02 having larger probabilities of non-detection than common taxa (0.2) with equivalent amplification efficiencies (colors). (TIFF)

## Author Contributions

**Conceptualization:** Zachary Gold, Andrew Olaf Shelton, Helen R. Casendino, Joe Duprey, Ramón Gallego, Amy Van Cise, Alexander J. Jensen, Erin D'Agnese, Elizabeth Andruszkiewicz Allan, Kim M. Parsons, Ryan P. Kelly.

**Data curation:** Zachary Gold, Andrew Olaf Shelton, Helen R. Casendino, Joe Duprey, Ramón Gallego, Amy Van Cise, Alexander J. Jensen, Michaela Labare, Kim M. Parsons, Ryan P. Kelly.

**Formal analysis:** Zachary Gold, Andrew Olaf Shelton, Helen R. Casendino, Joe Duprey, Ramón Gallego, Amy Van Cise, Mary Fisher, Alexander J. Jensen, Erin D'Agnese, Elizabeth Andruszkiewicz Allan, Ana Ramón-Laca, Kim M. Parsons, Ryan P. Kelly.

**Funding acquisition:** Zachary Gold, Andrew Olaf Shelton, Kim M. Parsons, Ryan P. Kelly.

**Investigation:** Zachary Gold, Andrew Olaf Shelton, Helen R. Casendino, Joe Duprey, Ramón Gallego, Amy Van Cise, Mary Fisher, Alexander J. Jensen, Erin D'Agnese, Elizabeth Andruszkiewicz Allan, Ana Ramón-Laca, Maya Garber-Yonts, Michaela Labare, Kim M. Parsons, Ryan P. Kelly.

**Methodology:** Zachary Gold, Andrew Olaf Shelton, Joe Duprey, Ramón Gallego, Amy Van Cise, Mary Fisher, Alexander J. Jensen, Erin D'Agnese, Elizabeth Andruszkiewicz Allan, Ana Ramón-Laca, Maya Garber-Yonts, Michaela Labare, Kim M. Parsons, Ryan P. Kelly.

**Project administration:** Zachary Gold, Andrew Olaf Shelton, Kim M. Parsons, Ryan P. Kelly.

**Resources:** Zachary Gold, Andrew Olaf Shelton, Alexander J. Jensen, Erin D'Agnese, Michaela Labare, Kim M. Parsons, Ryan P. Kelly.

**Software:** Zachary Gold, Andrew Olaf Shelton, Helen R. Casendino, Joe Duprey, Ramón Gallego, Amy Van Cise, Mary Fisher, Alexander J. Jensen, Erin D'Agnese, Elizabeth Andruszkiewicz Allan, Ana Ramón-Laca, Kim M. Parsons, Ryan P. Kelly.

**Supervision:** Zachary Gold, Andrew Olaf Shelton, Elizabeth Andruszkiewicz Allan, Kim M. Parsons, Ryan P. Kelly.

**Validation:** Zachary Gold, Andrew Olaf Shelton, Ramón Gallego, Alexander J. Jensen, Ana Ramón-Laca, Kim M. Parsons, Ryan P. Kelly.

**Visualization:** Zachary Gold, Andrew Olaf Shelton, Helen R. Casendino, Joe Duprey, Ramón Gallego, Amy Van Cise, Mary Fisher, Alexander J. Jensen, Erin D'Agnese, Elizabeth Andruszkiewicz Allan, Kim M. Parsons, Ryan P. Kelly.

**Writing – original draft:** Zachary Gold, Andrew Olaf Shelton, Kim M. Parsons.

**Writing – review & editing:** Zachary Gold, Andrew Olaf Shelton, Helen R. Casendino, Joe Duprey, Ramón Gallego, Amy Van Cise, Mary Fisher, Alexander J. Jensen, Erin D'Agnese, Elizabeth Andruszkiewicz Allan, Ana Ramón-Laca, Maya Garber-Yonts, Michaela Labare, Kim M. Parsons, Ryan P. Kelly.

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
