## [Decision Letter · Decision Letter 0]

6 Nov 2022

PONE-D-22-26685Distinguishing Signal from Noise: Understanding Patterns of Non-Detections to Inform Accurate Quantitative MetabarcodingPLOS ONE

Dear Dr. Gold,

Thank you for submitting your manuscript to PLOS ONE. After careful consideration, we feel that it has merit but does not fully meet PLOS ONE’s publication criteria as it currently stands. Therefore, we invite you to submit a revised version of the manuscript that addresses the points raised during the review process.

After a careful reading of the comments and concerns from the reviewers, I feel the manuscript needs a serious reformatting to accommodate the comments. Specifically, the consistency of the storyline and bridging between the cited articles, data and the underlying message of this manuscript. My understanding is that these comments will definitely help improve the status of the manuscript. 

Please submit your revised manuscript by Dec 21 2022 11:59PM. If you will need more time than this to complete your revisions, please reply to this message or contact the journal office at plosone@plos.org. Please include the following items when submitting your revised manuscript:A rebuttal letter that responds to each point raised by the academic editor and reviewer(s). You should upload this letter as a separate file labeled 'Response to Reviewers'.A marked-up copy of your manuscript that highlights changes made to the original version. You should upload this as a separate file labeled 'Revised Manuscript with Track Changes'.An unmarked version of your revised paper without tracked changes. You should upload this as a separate file labeled 'Manuscript'.

We look forward to receiving your revised manuscript.

Kind regards,

Tarunendu Mapder, Ph.D.

Academic Editor

PLOS ONE

Journal Requirements:

"ZG and AJJ were supported by the Joint Institute for the Study of the Atmosphere and Ocean (JISAO) under NOAA Cooperative Agreement NA15OAR4320063. RPK, EAA, and ED were supported by the Packard Foundation Grant (XX12345)."

"ZG and AJJ were supported by the Joint Institute for the Study of the Atmosphere and Ocean (JISAO) under NOAA Cooperative Agreement NA15OAR4320063. RPK, EAA, and ED were supported by the Packard Foundation Grant (XX12345)."

6. We noted in your submission details that a portion of your manuscript may have been presented or published elsewhere. [This manuscript has been posted as a pre-print on biorxiv: https://www.biorxiv.org/content/10.1101/2022.09.02.506420v1] Please clarify whether this [conference proceeding or publication] was peer-reviewed and formally published. If this work was previously peer-reviewed and published, in the cover letter please provide the reason that this work does not constitute dual publication and should be included in the current manuscript.

7. We note that you have stated that you will provide repository information for your data at acceptance. Should your manuscript be accepted for publication, we will hold it until you provide the relevant accession numbers or DOIs necessary to access your data. If you wish to make changes to your Data Availability statement, please describe these changes in your cover letter and we will update your Data Availability statement to reflect the information you provide.

Reviewers' comments:

Reviewer's Responses to Questions

**Comments to the Author**

1. Is the manuscript technically sound, and do the data support the conclusions?

Reviewer #1: Yes

Reviewer #2: Yes

2. Has the statistical analysis been performed appropriately and rigorously? 

Reviewer #1: Yes

Reviewer #2: Yes

3. Have the authors made all data underlying the findings in their manuscript fully available?

Reviewer #1: Yes

Reviewer #2: Yes

4. Is the manuscript presented in an intelligible fashion and written in standard English?

Reviewer #1: Yes

Reviewer #2: Yes

5. Review Comments to the Author

Reviewer #1: This is a very interesting paper!

I think this paper just needs some better links to the existing literature before publication.

# Comments on statistical simulation

Originally I thought simulation was unnecessary, since we could just compute $P(Y=0)$ directly. However, it's much harder than it first appears! I only got as far as an expression for $X_1$ (not even closed form!). I therefore agree with your approach to simulate but could you please link your work to some of the existing literature (below) on these distributions? And search the literature for anything relating to DNA amplification with regards to such distributions.

See attachment for the rest of the review.

Reviewer #2: Overall comments: The authors in this manuscript conduct a metabarcoding analysis of both mock communities and collected DNA samples to explore the factors that impact negatives in technical replicates. It was found that starting concentration and species-specific amplification efficiency. Overall, the methods are to be sound, and the presentation of the results is well thought out (the figures are also nicely done). However, I think the manuscript needs to be adjusted to better tell the story and highlight the relationships to the other papers that are commonly cited throughout. As it is currently written it comes across as though the results are not novel but just a repetition of previous work (see comments below for specifics).

Line 46: I think this paragraph could be flushed out a little more and include some more information to help set up the study and metabarcoding in general. With PLoS having such a broad audience it could be useful to just add a few sentences describing some specifics for how metabarcoding is rapidly advancing health, ecology, and conservation science. Possibly include a sentence on how metabarcoding results are more and more often influencing policy decisions, which would transition well into needing reliable estimates mentioned in the next paragraph.

Line 56-56: I think that this is area that could be a bit more specific on examples instead of just listing citations. Admittedly this is more of a stylistic/writing style preference and the information itself is sound, I just think it would make it a little more gripping to read.

Line 66: This could be a great spot for a summary figure highlighting the compounding processes across the DNA collection and analysis pipeline. Including the aspects you do not focus on such as the ones mentioned in the discussion or bioinformatic or sequencing platforms (see comment below).

Line 127: Does this species-specific amplification efficiency that influences the amplicons expand to genus and family level biases that can occur simultaneously in the process? For instance, if a specific species has species-specific biases but is also part of a genus that is historically biased would that be included?

Line 221: Would GBLOCK sequences have been more useful here to ensure no nDNA or other sources got through your nested PCR strategy? Obviously you cannot go back and change anything but I am curious if this would have been more effective or not.

Line 238: Why did you have to resequence, I’m assuming the run failed due to a technical failure as is so common, but was it something that could be affecting the other runs as well to a lesser degree. I am doubtful, but it may be best to be clear why the resequencing took place.

Line 261: These hypotheses could be expanded on a bit or shifted to be a little more novel. The idea that fewer non-detections for more abundant DNA molecules seems a little obvious the way it is currently written. I agree it’s an important question to get at, but it should be reworded a bit. If this cannot be reworded, I would recommend adding a bit more text to the intro to highlight why this may not be as obvious as one would expect. This comment also relates to my comments down below about framing the story of this manuscript around the previous work.

Line 303 and 197: I think having read this results section there needs to be a bit more information on these samples, what they were, did they have multiple species within them etc. I know it is also in Gold et al. but it would be helpful to repeat some of this information here. Additionally, it is a little confusing teasing out these 84 ethanol-preserved samples and the formalin preserved larvae? Same as above, where more information about the previous work in other manuscripts would help make this clearer.

Line 303: Something that I would be interested to hear the authors point of view on is whether or not these samples actually constitute eDNA. Typically, eDNA is referred to as bulk non-targeted samples but someone could argue that larvae within an ethanol bottle would not be considered eDNA sampling. This is a topic that can be rather contentious based on the audience (similar debates take place over fecal sampling), so I am curious to hear the author’s take. It is also possible (related to my above comment) that I am not clear on what these samples actually are.

Line 394: I think this illustrates my comment above about your first hypothesis seemingly already being well established across the literature.

Line 467: In addition to processes from the start of eDNA collection, it could be useful to mention the impacts bioinformatic pipeline choice and sequencer selection can have on the results as well.

Line 507: I apologize if I missed this, but could there be a little bit of discussion about the limitations that may arise within your analysis from only having 2 species with lower amplification efficiency. This is a fairly small sample size, especially compared to the mock community that had 15 species with lower efficiency.

Line 514: I apologize in advance for the length of this comment but overall, it is not completely clear the relationship and story that is being told across the various manuscripts, and as a result, it makes this work seem less novel comparatively. To my understanding….Gold (31) is where the initial samples came from, and you are reusing the samples. It also seems like these results are from the same analysis from Gold 31(?). While the Shelton et al. (27) paper seems to have initially designed the model and tested it with mock communities. Now this current paper, is using the samples from Gold 31 and testing the models described with Shelton. Though it looks like the Shelton paper has already done some mock community testing which is repeated in this reviewed paper? Assuming my assessment is correct, this took a good bit of searching and reasoning to put together, which is not ideal for a typical reader. I know you mention occasionally in the text that certain aspects come from Shelton or Gold, but the relationship is still not made very clear. For example, the ending sentence says that “together with Shelton et al.” but again that relationship across these manuscripts is unclear and as readers we didn’t know these results were being combined with Shelton from the beginning. I do believe the information presented here is novel, but the story in the introduction needs to be told in a way that highlights the path this data is taking. For example, explicitly stating that this work is a direct continuation of Shelton et al.’s model and telling the story in the introduction to guide the reader who may not know these works. Also, its mentioned that the mock communities are from Shelton et al. did they also calculate the efficiencies for these same species or was that new to this manuscript as it is not clear how it is written? If these were already calculated in Shelton et al. it may not need to be repeated to such degree here.

6. PLOS authors have the option to publish the peer review history of their article (what does this mean?). If published, this will include your full peer review and any attached files.

Reviewer #1: No

Reviewer #2: No

---

## [Author Response · Author response to Decision Letter 0]

12 Jan 2023

Please see attached response to reviewer document.

---

## [Decision Letter · Decision Letter 1]

28 Apr 2023

Signal and Noise in Metabarcoding Data

PONE-D-22-26685R1

Dear Dr. Gold,

We’re pleased to inform you that your manuscript has been judged scientifically suitable for publication and will be formally accepted for publication once it meets all outstanding technical requirements.

Kind regards,

Ruslan Kalendar

Academic Editor

PLOS ONE

Reviewers' comments:

Reviewer's Responses to Questions

**Comments to the Author**

Reviewer #2: 

I appreciate and thank the authors for the thought and effort that went in to addressing my comments. I enjoyed the manuscript and feel that the manuscript is improved and tells a very interesting story.

I only have an extremely minor comment..

Line 64-65: change characterizing to characterize

---

## [Editor Report · Acceptance letter]

3 May 2023

PONE-D-22-26685R1 

Signal and Noise in Metabarcoding Data 

Dear Dr. Gold:

I'm pleased to inform you that your manuscript has been deemed suitable for publication in PLOS ONE. Congratulations! Your manuscript is now with our production department. 

Kind regards, 

on behalf of

Professor Ruslan Kalendar 

Academic Editor

PLOS ONE